# Predictors of Adverse Cardiovascular Events After CABG in Patients with Previous Heart Failure

**DOI:** 10.3390/life15030387

**Published:** 2025-02-28

**Authors:** Alla Garganeeva, Elena Kuzheleva, Olga Tukish, Michail Kondratiev, Karina Vitt, Sergey Andreev, Yury Bogdanov, Oksana Ogurkova

**Affiliations:** Cardiology Research Institute, Tomsk National Research Medical Center, Russian Academy of Sciences, Tomsk 634055, Russia; aag@cardio-tomsk.ru (A.G.); olgatukish@yandex.ru (O.T.); kmu@cardio-tomsk.ru (M.K.); karinavitt@list.ru (K.V.); anselen@rambler.ru (S.A.); yuri-bogdanov@mail.ru (Y.B.); oon@cardio-tomsk.ru (O.O.)

**Keywords:** heart failure, CABG, GDF-15, medication adherence

## Abstract

Coronary artery disease (CAD) is the primary risk factor for heart failure (HF) development. Coronary artery bypass graft (CABG) surgery remains the gold-standard treatment for multivessel coronary artery disease. The purpose of this study was to identify predictors of cardiovascular events in patients after CABG by looking at clinical parameters, examining biomarkers of inflammation and fibrosis, and assessing patients’ adherence to heart failure therapy before CABG. The prospective observational study included consecutively hospitalized patients with HF and CAD eligible for CABG (*n* = 82). The study’s primary endpoint was a combination (MACE) of cardiac death, hospitalization with HF, acute ischemic events requiring unplanned revascularization, or stroke. The enzyme-linked immunosorbent assay was performed to assess serum levels of NGAL, GDF-15, NTproBNP, TGF beta, and hsCRP. The participants’ medication adherence level was assessed using the Morisky–Green scale. A total of 37 events were registered (45.1%) at follow-up (36 (26; 43) months). All patients were divided into two groups: Group 1 (*n* = 45) comprised patients without the combined endpoint, and Group 2 (*n* = 37) comprised patient who suffered adverse cardiovascular events. A high GDF-15 level and low adherence based on the Morisky–Green scale were independent predictors of a MACE at follow-up. The median time before the development of the MACE which was predicted based on Kaplan–Meier analysis in the group with a GDF-15 value less than 2064 pg/mL was 64 (50; 80) months, and in the group with a GDF-15 value more than or equal to 2064 pg/mL, it was 40 (34; 46) months (*p* < 0.001). Higher GDF-15 values and poor adherence to treatment are associated with adverse cardiovascular events in patients with HF and CAD who have undergone CABG. However, further studies are needed to support the use of GDF-15 as a prognostic marker in real-life clinical practice.

## 1. Introduction

Coronary artery disease (CAD) is the primary risk factor for heart failure (HF) development [1]. About half of patients with acute and chronic HF have an ischemic etiology [2]. Coronary artery bypass graft (CABG) surgery remains the gold-standard treatment for multivessel coronary artery disease. CABG results in a significant reduction in major adverse cardiac and cerebrovascular events (MACEs) [3,4]. However, available data suggest that up to 40–50% of patients may experience treatment failure within 10 years after CABG [3,5]. Patients undergoing CABG remain at risk for recurrent adverse cardiovascular outcomes (myocardial infarction (MI), stroke, cardiac death, HF hospitalization) [6]. Factors associated with the development of adverse cardiovascular events, including cardiac death, may include old age, the location of residence, ejection fraction, a history of stroke [7,8], diabetes and obesity, and chronic kidney disease, among others [9,10]. Drug therapy for coronary heart disease and heart failure, as well as comorbid diseases, reduces the risk of an unfavorable prognosis. At the same time, poor adherence and persistence (duration of time for which medications and healthy behaviors are continued) have a profound effect on effective management, patient safety, and outcomes [2]. Currently, a search is underway for preoperative predictors of adverse outcomes after CABG, including social factors [11], clinical data, laboratory data (NTproBNP [12], ST2, Galectin-3, etc.) [13,14], and instrumental data [8,15], that can be used for risk stratification or estimating postsurgical prognosis. Identified modifiable factors can be modeled to assist in implementing proactive measures for the prevention of cardiovascular events after surgery [16,17].

The purpose of this study was to identify predictors of adverse cardiovascular events after CABG in patients with previous heart failure by looking at clinical data, standard laboratory and instrumental data, biomarkers (NGAL, GDF-15, NTproBNP, TGF beta 1, hsCRP), and patients’ adherence before CABG.

## 2. Materials and Methods

This prospective observational study included consecutively hospitalized patients with HF who underwent CABG (inclusion period: 2018–2020).

Inclusion criteria: HF, multivessel atherosclerosis of the coronary arteries (triple vessel disease), planned CABG by the decision of the Heart Team, and patient consent to the intervention and to participation in the study.

Non-inclusion criteria: refusal to participate in the study, myocardial infarction (MI) within the last 6 months, stroke within the last 6 months, implanted cardiac rhythm management devices, the need for additional cardiac surgery, except CAGB, or the presence of advanced kidney disease (eGFR < 30 mL/min/1.73 m^2^) or severe related diseases (oncological diseases in the active stage, infiltrating heart diseases, autoimmune diseases, acute infectious, and exacerbation of chronic somatic diseases).

The diagnosis of HF was based on the current clinical guidelines [1]. The multivascular stenosing atherosclerosis of coronary arteries was diagnosed using invasive coronary angiography on the angiographic complex “Siemens Artis One” and the ACOM.PC of Siemens (Forchheim, Germany) based on clinical indications. Echocardiography was conducted with the “Philips HD15 Ultrasound system”.

The research protocol adhered to the principles of the Declaration of Helsinki and was approved by the Local Ethics Committee (Protocol No. 188 of 18.09.2019). Before any procedure, all patients had signed a form of voluntary informed consent.

Anamnesis, physical examination, and standard laboratory and instrumental examinations were performed 3–5 days before surgery.

### 2.1. Laboratory Examination

Blood was collected one day before CABG from the cubital vein in the morning on an empty stomach. Further preparation of blood samples for analysis included centrifuging, serum separation, and freezing at −80 °C. The analysis was conducted after one round of blood serum thawing.

The enzyme-linked immunosorbent assay was performed to assess serum levels of neutrophil gelatinase-associated lipocalin (NGAL) (ng/mL) (BioVendor, Brno, Czech Republic), growth/differentiation factor 15 (GDF-15) (pg/mL) (BioVendor, Czech Republic), NTproBNP (pg/mL), (Biomedica GmbH, Vienna, Austria), transforming growth factor beta 1 (TGF beta 1) (pg/mL) (Thermo Fisher Scientific Inc., Waltham, MA, USA), and high-sensitivity C-reactive protein (mg/L) (hsCRP) (Biomerica C-reactive protein, Irvine, CA, USA). The research was carried out using the equipment of the Center for Collective Use “Medical Genomics” of Cardiology Research Institute, Tomsk National Research Medical Center, Russian Academy of Sciences, Tomsk, Russia.

### 2.2. Assessment of Adherence

The participants’ medication adherence level was assessed using the Morisky–Green scale [6]. The patients were considered adherent if they scored ≥ 3 pts. on the Morisky–Green scale.

Endpoints and follow-up: the primary endpoint of this study was a major adverse cardiac event (MACE), defined as cardiovascular death, acute ischemic events requiring unplanned revascularization, HF hospitalization, or stroke.

### 2.3. Statistical Analysis

We planned to evaluate the following predictors: the NYHA class of HF, LVEF, glomerular filtration rate, adherence to treatment, GDF-15, NGAL, TGF beta 1, and hsCRP. Data were analyzed with the use of the program “IBM SPSS 21”. Continuous variables are presented as the median and interquartile range (Me (Q25; Q75)), considering the non-normal distribution of parameters. Categorical data are presented in absolute and relative values: *n* (%). Continuous variables in the independent samples were analyzed using the Mann–Whitney criterion. Spearman’s rank coefficient of correlation was used to estimate correlation relationships. The statistical significance of differences for categorical variables was determined using the χ^2^ Pearson criterion and two-sided Fisher’s exact test. The influence of the studied factors on the development of a MACE was analyzed using logistic regression analysis and receiver operating characteristic (ROC) analysis (calculated area under the curve (AUC)). The Kaplan–Meier hazard function was used to estimate the time to endpoint, and Log-rank was used to compare hazard curves. The risk of developing a combined endpoint depending on the level of GDF-15, taking into account interfering factors, was assessed using Cox regression. A value of *p* < 0.05 was considered statistically significant.

## 3. Results

### 3.1. Patients’ Characteristics

In this study, 82 patients with CAD who underwent CABG were enrolled. The median age was 63 (58; 68) years, and 89% of patients were male. The median body mass index (BMI) was 27.8 (25.4; 31.4) kg/m^2^, and 69.5% of patients had a family history of CAD. Nearly half (56%) of the patients reported a smoking history. The median of the LVEF was 49.5 (35; 63)%, and the NTproBNP value was 223 (179; 471) pg/mL. A total of 15 patients (18.3%) had left coronary artery trunk stenosis, 73 patients (88%) had stenosis of the anterior descending artery, 46 patients (56%) had circumflex artery stenosis, and 61 patients (74%) had right coronary artery stenosis. The median and interquartile range of the SYNTAX score were 24 (17.75; 30).

### 3.2. Outcomes

The median follow-up period was 36 (26; 43) months. A total of 37 events were registered (45.1%) at follow-up. Death from cardiac causes was registered in 13 cases (15.9%). Hospitalization for decompensated heart failure was registered in 12 patients (14.6%), urgent revascularization or myocardial infarction in 10 patients (12.2%), and stroke in 2 patients (2.4%). The median time to the development of the endpoint was 26 (13.42) months. After the follow-up period, all patients were divided into two groups: Group 1 (*n* = 45) comprised patients without a combined endpoint, and Group 2 (*n* = 37) comprised patient who suffered adverse cardiovascular events.

### 3.3. Subgroup Comparison

The main characteristics of the study groups are presented in Table 1. Differences in sex, age, body mass index, history of hypertension, diabetes, and comorbidity between groups were found to be negligible. Some clinical symptoms and signs of HF were more pronounced in Group 2 compared to Group 1. The proportions of patients with edema and NYHA III were higher in Group 2 (*p* < 0.05). A history of myocardial infarction was registered more often in Group 2.

GDF-15 was the only inflammation marker that differed between groups: patients who experienced a MACE had a higher concentration of this marker (<0.001). The values of NGAL, TGF beta1, C-reactive protein, and the neutrophil-to-lymphocyte ratio were similar in both groups. The NT-proBNP value was not different between groups and met the diagnosis of heart failure (Table 2).

The baseline echocardiographic parameters were different in the two groups (Appendix A). The LVEF was significantly lower (*p* < 0.001) in Group 2 (60 (44; 64) % and 37 (28.5; 60) % in Group 1 and Group 2, respectively). End-systolic volume, end-diastolic volume, indexed to body surface area, and the degree of mitral and tricuspid regurgitation were much larger in Group 2 (*p* < 0.05). Zones of hypo- or akinesia were more often diagnosed in this group too (*p* = 0.008). Thus, preoperative echocardiographic parameter values were significantly worse in patients with a MACE in the follow-up period.

Coronary atherosclerosis severity was similar in the groups: the median and interquartile range of the SYNTAX score were 24 (19; 30) in Group 1 and 22.5 (16.5; 30) in Group 2 (*p* = 0.450) (Table 3). On-pump CABG was completed by all patients. The number of coronary grafts and the technique parameters of CABG were comparable in the groups: most patients required three coronary grafts, and cardiopulmonary bypass time (105 (84.5; 118) and 110 (78.5; 195.5) min, *p* = 0.532) and aortic cross-clamping time (60 (51.5; 74) and 70.5 (54; 140.75) min, *p* = 0.327) were comparable in the groups. The number of complications and duration of hospitalization were similar in the two groups (*p* > 0.05).

Anemia at discharge was more often diagnosed in Group 2, but these differences were not statistically significant (46.7% in Group 1 and 67.6% in Group 2, *p* = 0.058).

Drug therapy before hospitalization was comparable in the two groups, excluding mineralocorticoid receptor antagonists (4.4% and 27%, *p* = 0.004), which were more often required by patients of Group 2. Medical therapy after discharge was comparable in the two groups (Table 4). The level of treatment adherence was lower in Group 2 based on the Morisky–Green scale: 28.9% of patients in Group 1 and 75.7% of patients in Group 2 were not adherent to treatment (*p* < 0.001).

### 3.4. Predictors of Outcomes

Parameters that were significantly different between groups in univariate analysis were included in the multivariate logistic regression and Cox regression analysis. There was a correlation between most parameters of transthoracic echocardiography, and we included only the left ventricular ejection fraction parameter in the regression analysis. NTproBNP was also added to the multivariate analysis given its high clinical significance in patients with heart failure after CABG. The results of multivariate logistic regression analysis are presented in Table 5. A high GDF-15 level and low adherence based on the Morisky–Green score were independent predictors of a MACE at follow-up. The statistical significance of the model is high: chi-square = 39.986, *p* < 0.001.

Based on receiver operating characteristic analysis, the value obtained in the regression equation was prognostically significant, and cut-off values predicting the development of MACEs after CABG were identified. The value of ≥0.26 was a predictor for MACEs during the 36 months after CABG (the area under the ROC curve was 0.871, with a sensitivity of 92% and a specificity of 67%) (Figure 1).

The receiver operating characteristic analysis was also carried out for the GDF-15 value (Figure 2). The value of ≥2064 pg/mL was a predictor for MACEs during the 36 months after CABG (the area under the ROC curve was 0.738, with a sensitivity of 84.5% and a specificity of 55.6%).

We divided patients into two groups depending on their GDF-15 values (<2064 pg/mL and ≥2064 pg/mL). The Kaplan–Meier curves diverged significantly (Log-rank, *p* < 0.001) (Figure 3). The median time before the development of the MACE which was predicted based on the Kaplan–Meier analysis in the group with a GDF-15 value less than 2064 pg/mL was 64 (50; 80) months, and in the group with a GDF-15 value more than or equal to 2064 pg/mL, it was 40 (34; 46) months (*p* < 0.001).

Cox regression was performed taking into account all covariates that were included in the logistic regression analysis and that demonstrated statistically significant differences in the groups in the univariate analysis. A GDF-15 value of more than 2064 pg/mL was an independent predictor of the development of the combined endpoint (OR 7.220 95%CI 2.398–21.739) (Figure 4, Table 6).

## 4. Discussion

In our study, a high incidence of MACEs during the 36 months after CABG was recorded—45.1%, including cardiac death, hospitalization with HF, acute ischemic events requiring unplanned revascularization, or stroke. This is comparable to the results of the STIHC trial: death from any cause or hospitalization due to cardiovascular causes was registered in 45% of cases in 3 years of follow-up [18]. But the STICH study included patients with a reduced left ventricular ejection fraction (≤35%), whereas in our group, the proportion of such patients was only 28%.

The rate of adverse cardiovascular events was 22.6% in 36 months of follow-up after CABG in the SYNTAX trial [19], 12% in the PRECOMBAT trial [20], and 23.7% in the NOBLE trial [21]. But these studies included patients without heart failure in most cases. Considering that our follow-up period coincided with the COVID-19 pandemic, we cannot exclude the influence of viral infection on our results.

Predictors of adverse cardiovascular events within 3 years after CABG in our study were bilateral leg edema, a history of MI, NYHA class, treatment adherence as measured by Morisky–Green scores, total cholesterol, triglycerides, GDF-15, the use of mineralocorticoid receptor antagonists before hospitalization, and echocardiographic parameters obtained from patients before CABG. Most of these parameters have demonstrated their prognostic significance after CABG in previous studies [7,22,23]. In our study, NTproBNP was not statistically significantly associated with the prognosis of patients, which is not consistent with other studies [12,13]. This is most likely due to the fact that in our cohort, the median NTproBNP values were 167.4 pg/mL in patients without MACEs and 326.7 pg/mL in patients with MACEs, whereas according to previous studies, higher NTproBNP values (mainly more than 400 pg/mL) were associated with MACEs [12]. However, the concentration of NTproBNP was significantly higher in the group of patients who suffered adverse cardiovascular events. Therefore, we included this parameter in the multivariate analysis but did not obtain a statistically significant result.

Independent predictors of adverse cardiovascular events within 3 years after CABG in our study were poor adherence to treatment and a GDF-15 value greater than or equal to 2064 pg/mL.

GDF-15 is associated with CAD severity [24,25,26]. In patients with CAD, the levels of GDF-15 at admission were associated with an increased 1-year risk of CV death, HF, and bleeding outcomes [27,28].

Previous studies have reported increased circulating concentrations of GDF-15 in HF, suggesting the potential prognostic significance of GDF-15 in this setting. A meta-analysis of 10 studies involving 6244 patients with HF showed the association of GDF-15 with the risk of all-cause mortality among patients with chronic ischemic HF (HR: 1.75, 95%CI: 1.24–2.48, *p* = 0.002), while this association was not found among patients with chronic nonischemic HF (HR: 1.01, 95%CI: 1.00–1.02, *p* = 0.219) [29].

Higher levels of GDF-15 were associated with an increased risk of cardiovascular death, hospitalizations for heart failure, and worse kidney outcomes according to an analysis in the cohort of patients included in the EMPEROR-Reduced and EMPEROR-Preserved studies (n = 1124) [30].

Thus, we expected to identify an association between GDF-15 and the prognosis of patients with HF and CAD. However, in the available publications, there are no data on the association between the concentration of GDF-15 and the development of adverse cardiovascular events in such patients after CABG.

In this study, we have also shown for the first time that an elevated GDF-15 concentration is an independent risk factor for adverse cardiovascular events within 36 months after CABG in patients with CAD and HF. The mechanism of the biological action of HDF 15 has not been studied to date. An increase in its concentration is associated with inflammation, fibrosis, the aging process, and metabolic disorders [31]. It is now speculated that circulating GDF-15 levels might reflect mitochondrial dysfunction [32]. It is highly likely that any one of these mechanisms or their combination is involved in the progression of coronary and heart failure after CABG, which links this marker to the prognosis of patients. The concentration of GDF-15 ≥ 2064 pg/mL was an independent predictor of the combined endpoint, according to the results of Cox regression analysis. Considering the data we obtained, as well as additional data from a recent study in the EMPEROR-Preserved and EMPEROR-Reduced cohorts [30], optimal drug therapy for CHF may be particularly beneficial in patients with elevated levels of GDF-15 prior to surgical myocardial revascularization. However, this assumption requires further investigation.

The second independent predictor of MACEs in our study was poor patient adherence. To determine adherence, we used the Morisky–Green scale, consisting of four questions. According to the results of logistic regression, each additional point on the Morisky–Green questionnaire significantly reduces the likelihood of experiencing adverse cardiovascular events (*p* = 0.005). In patients with cardiovascular diseases, the rate of poor adherence has remained stable over two decades: 40% of patients fail to fill an original prescription and over 50% discontinue medications within a year, which results in a considerable gap between recommendations and actual clinical practice [33]. This result once again highlights the need to work with patients with the aim of increasing adherence to treatment.

### Study Limitations

The main limitation of this study was the small sample size, but an advantage is that there were no patients lost to follow-up. Another limitation is the single-center nature of the study and the COVID-19 pandemic, which occurred during the follow-up period. However, the data from this work add to data on the important prognostic role of GDF-15 in patients with heart failure.

## 5. Conclusions

Higher GDF-15 values and poor adherence to treatment were independent predictors of adverse cardiovascular events in patients with HF and CAD who had undergone CABG. However, further studies are needed to support the use of GDF-15 as a prognostic marker in real-life clinical practice. Our preliminary results should be corroborated in larger cohorts.

## Figures and Tables

**Figure 1 life-15-00387-f001:**
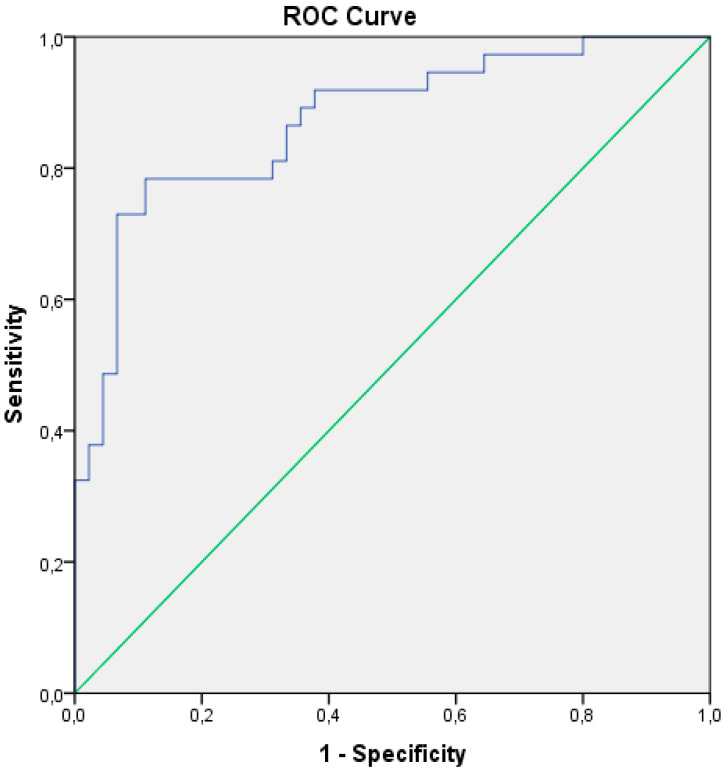
ROC curve of the value obtained in the regression equation for MACE prediction. The value of ≥0.26 was a predictor for MACEs during the 36 months after CABG (the area under the ROC curve was 0.867, with a sensitivity of 92% and a specificity of 67%). MACE—major acute cardiac event; CABG—coronary artery bypass grafting; ROC—receiver operating characteristic.

**Figure 2 life-15-00387-f002:**
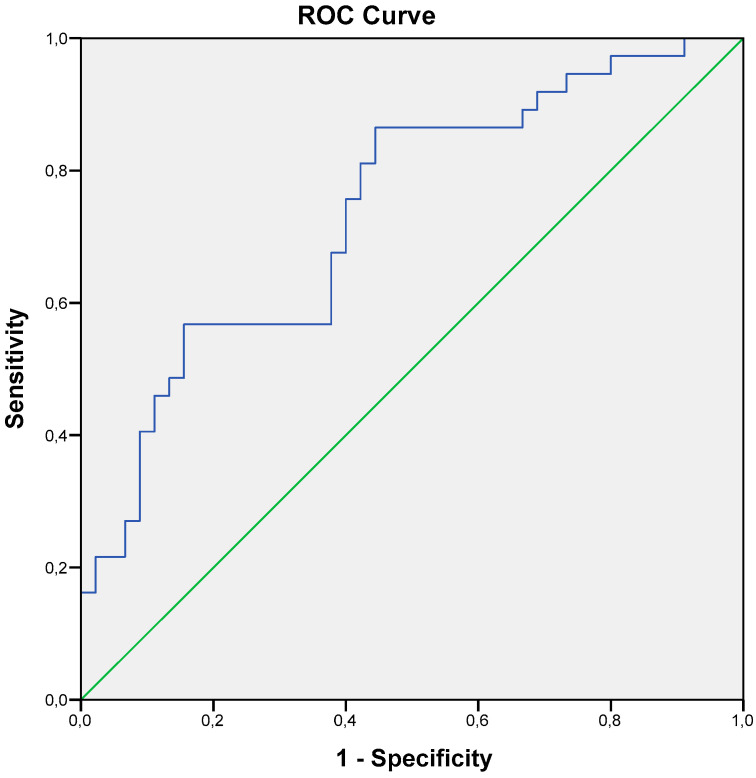
ROC curve of GDF-15 value for MACE prediction. The value of ≥2064 pg/mL was a predictor for MACEs during the 36 months after CABG (the area under the ROC curve was 0.738, with a sensitivity of 84.5% and a specificity of 55.6%). MACE—major acute cardiac event; CABG—coronary artery bypass grafting; ROC—receiver operating characteristic.

**Figure 3 life-15-00387-f003:**
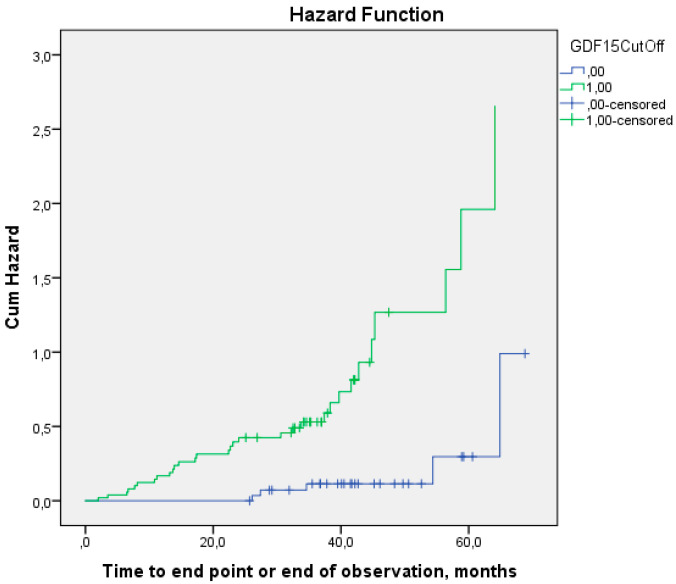
Kaplan–Meier hazard function before the development of the MACE in patients after CABG. 0—group with GDF-15 value less than 2064; 1—group with GDF-15 value more than or equal to 2064 (*p* < 0.001).

**Figure 4 life-15-00387-f004:**
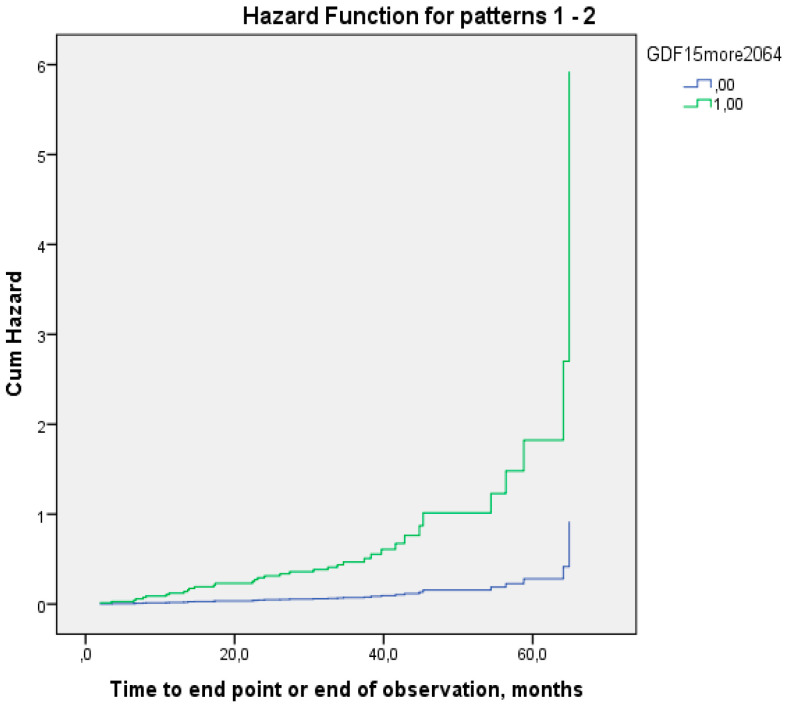
Development of combined endpoint in patients with GADF15 levels < 2064 pg/mL (1) and ≥2064 pg/mL (2) with edema, history of MI, NYHA class, adherence to treatment on Morisky–Green scale, total cholesterol, triglycerides, LVEF, mineralocorticoid receptor antagonists before hospitalization, and NTproBNP. OR 7.22 95%CI 2.398–20.739 (*p* < 0.001).

**Table 1 life-15-00387-t001:** Baseline demographic and clinical characteristics of patients before CABG.

Parameters	Group 1 (*n* = 45)	Group 2 (*n* = 37)	*p* Value
Age; years	61 (56; 68)	63 (60; 68)	0.341
Gender: male; *n* (%)	40 (88.9)	33 (89.2)	0.965
Social status: alone; *n* (%)	2 (4.4)	6 (16.2)	0.074
Angina; *n* (%)	45 (100)	37 (100)	1.000
Dyspnea; *n* (%): 0—none 1—with significant physical activity 2—with normal activity 3—at rest 4—in horizontal position	0—3 (6.7); 1—11 (24.4); 2—31 (68.9); 3—0 4—0	0—4 (10.8) 1—5 (13.5); 2—26 (70.3) 3—1 (2.7) 4—1 (2.7)	0.394
Heart palpitations; *n* (%)	11 (24.4)	14 (37.8)	0.190
Bilateral leg edema; *n* (%)	5 (11.1)	11 (29.7)	0.034
Weakness; *n* (%)	21 (46.5)	17 (45.9)	0.948
Liver enlargement; *n* (%)	1 (2.2)	0	1.000
Moist rales; *n* (%)	0	3 (8.1)	0.088
Lungs congestion by X-ray	7 (15.6)	11 (29.7)	0.123
Duration of history of coronary artery disease; years	3 (1; 10)	4 (1.25; 12)	0.452
History of myocardial infarction; *n* (%)	26 (57.8)	29 (78.4)	0.048
Two and more myocardial infarctions in history	4 (8.9)	8 (21.6)	0.105
Angina pectoris functional class	3 (2; 3)	3 (2; 3)	0.664
NYHA class	2 (2; 3)	3 (2; 3)	0.007
Hypertension; *n* (%)	44 (97.8)	37 (100)	0.106
Duration of history of hypertension; years	10 (6.5; 20)	10 (6; 17)	0.271
Stroke; *n* (%)	0	3 (8.1)	0.088
Diabetes; *n* (%)	11 (24.4)	9 (24.3)	0.990
Impaired glucose tolerance or DM2; *n* (%)	13 (28.9)	9 (24.3)	0.642
Peripheral atherosclerosis ≥ 40%; *n* (%)	13 (28.9)	13 (35.1)	0.545
Atrial fibrillation; *n* (%)	9 (20)	11 (29.7)	0.307
Type of AF: 0—no; 1—paroxysmal; 2—persistent; 3—long-term persistent; 4—permanent	1—4 (8.9); 2—2 (4.4); 3—1 (2.2); 4—2 (4.4)	1—4 (10.8); 2—3 (8.1); 3—1 (2.7); 4—3 (8.1)	0.871
Ventricular extrasystole grade III-V by Lown–Wolf; *n* (%)	7 (15.6)	12 (32.4)	0.060
Family history of CVD; *n* (%)	35 (77.8)	22 (59.5)	0.073
Chronic obstructive pulmonary disease; *n* (%)	8 (17.8)	9 (24.3)	0.467
Stomach ulcer; *n* (%)	11 (24.4)	14 (37.8)	0.190
Current smoker; *n* (%)	27 (60)	19 (51.4)	0.432
Weight; kg	81 (73; 91.5)	85 (76; 92)	0.310
Body mass index; kg/m^2^	27.8 (22.35; 31.25)	28.4 (25.35; 32.35)	0.748
Overweight or obesity; *n* (%)	36 (80)	30 (81.1)	0.902
Obesity; *n* (%)	16 (35.6)	13 (35.1)	0.968
Waist circumference; cm	95 (87; 101)	94 (81; 101.5)	0.473
Systolic blood pressure; mm Hg	124 (117.5; 130)	125 (120; 130)	0.583
Diastolic blood pressure; mm Hg	80 (70; 80)	80 (70; 81)	0.181
Heart rate; b.p.m	67 (64; 72.75)	72 (64; 80)	0.069

DM2—diabetes type 2; AF—atrial fibrillation; CVD—cardiovascular diseases. Continuous variables are presented as median and interquartile range (Me (Q25; Q75)). Categorical data are presented in absolute and relative values: *n* (%).

**Table 2 life-15-00387-t002:** Laboratory data of patients before CABG.

Parameters	Group 1 (*n* = 45)	Group 2 (*n* = 37)	*p* Value
Hemoglobin, g/dL	14.9 (13.95; 16.05)	14.3 (12.95; 15.7)	0.109
Creatinine, mg/dL	1.1 (0.95; 1.27)	1.16 (1; 1.27)	0.428
eGFR, mL/min/m^3^	73 (59; 81.5)	69 (58; 77)	0.250
Fasting plasma glucose, mmol/L	5.6 (5.2; 6.54)	5.51 (5.08; 5.91)	0.292
Total cholesterol, mmol/L	4.25 (3.72; 5.79)	4.02 (3.19; 5)	0.044
Triglycerides, mmol/L	1.72 (1.18; 2.23)	1.21 (0.93; 1.81)	0.038
Low-density lipoprotein, mmol/L	2.42 (2.09; 3.77)	2.37 (1.41; 3.04)	0.096
High-density lipoprotein, mmol/L	1.09 (0.94; 1.2)	1.07 (0.9; 1.28)	0.819
NGAL, ng/mLmL	41.3 (33.6; 55.4)	34 (25.7; 52)	0.070
GDF-15, pg/mL	1997 (1469.5; 2384)	2590 (2144.25; 3733)	<0.001
NT-proBNP, pg/mL	167.4 (113.85; 422.25)	326.7 (139; 590)	0.066
TGF beta1, pg/mLmL	57,600 (44,490; 69,675)	57,510 (45,725; 75,135)	0.776
CRP, mg/L	4.7 (2.2; 8.6)	4.9 (2.1; 9.2)	0.453

GFR—glomerular filtration rate by CKD-EPI; NGAL—neutrophil gelatinase-associated lipocalin; GDF-15—growth/differentiation factor 15; TGF beta1—transforming growth factor beta 1; CRP—high-sensitivity C-reactive protein. Continuous variables are presented as median and interquartile range (Me (Q25; Q75). Categorical data are presented in absolute and relative values: n (%).

**Table 3 life-15-00387-t003:** Coronary atherosclerosis severity and CABG specification in study groups.

Parameters	Group 1 (*n* = 45)	Group 2 (*n* = 37)	*p* Value
Stenosis of the anterior descending artery	40 (88.9)	32 (86.5)	0.749
Left coronary artery trunk stenosis: 0—no; 1—yes	7 (15.6)	8 (21.6)	0.480
Stenosis of the right coronary artery: 0—no; 1—yes	33 (73.3)	28 (75.5)	0.809
Circumflex artery stenosis: 0—no; 1—yes	25 (55.6)	21 (56.8)	0.913
SYNTAX score	24 (19; 30)	22.5 (16.5; 30)	0.450
Number of coronary bypass grafts	3 (2;3)	3 (2;3)	0.330
Complications; *n* (%)	24 (53.3)	24 (64.9)	0.292
Infectious complications; *n* (%)	10 (22.2)	6 (16.2)	0.495
Postpericardiotomy syndrome; *n* (%)	21 (46.7)	20 (54.9)	0.506
Complications: cardiac arrhythmias; *n* (%)	3 (6.7)	5 (13.5)	0.535
Fluid in pleural cavities or pericardium at discharge; *n* (%)	11 (24.4)	10 (27)	0.79
Anemia at discharge; *n* (%)	21 (46.7)	25 (67.6)	0.058
Duration of hospitalization; bed days	22 (20; 27.5)	22 (19; 26.5)	0.918

Continuous variables are presented as median and interquartile range (Me (Q25; Q75). Categorical data are presented in absolute and relative values: n (%).

**Table 4 life-15-00387-t004:** The medical therapies in study groups.

Parameters	Group 1 (*n* = 45)	Group 2 (*n* = 37)	*p* Value
**Adherence to treatment (Morisky–Green scores)**	**3 (2; ** **4)**	**2 (1; ** **2.5)**	<0.001
**Drug therapy before hospitalization**			
Nitrates before hospitalization; *n* (%)	12 (26.7)	7 (18.9)	0.408
Beta-blockers before hospitalization; *n* (%)	32 (71.1)	24 (64.9)	0.545
Acetylsalicylic acid before hospitalization; *n* (%)	28 (62.2)	21 (56.8)	0.616
Clopidogrel before hospitalization; *n* (%)	15 (33.3)	6 (16.2)	0.077
Dual antiplatelet therapy; *n* (%)	13 (28.9)	5 (13.5)	0.094
Warfarin before hospitalization; *n* (%)	2 (4.4)	3 (8.1)	0.49
Direct oral anticoagulants before hospitalization; *n* (%)	3 (6.7)	4 (10.8)	0.504
ACE inhibitor before hospitalization; *n* (%)	16 (35.6)	12 (32.4)	0.767
Angiotensin receptor blockers before hospitalization; *n* (%)	13 (28.9)	9 (24.3)	0.642
Any RAAS blocker before hospitalization; *n* (%)	29 (64.4)	21 (56.8)	0.478
Statins before hospitalization; *n* (%)	33 (73.3)	22 (59.5)	0.183
Mineralocorticoid receptor antagonists before hospitalization; *n* (%)	2 (4.4)	10 (27)	0.004
Loop diuretics before hospitalization; *n* (%)	16 (35.6)	21 (56.8)	0.055
**Drug therapy after discharge**			
Nitrates; *n* (%)	2 (4.4)	3 (8.1)	0.654
Beta-blockers; *n* (%)	39 (86.7)	28 (75.7)	0.2
Acetylsalicylic acid; *n* (%)	42 (93.3)	31 (83.8)	0.169
Clopidogrel; *n* (%)	29 (64.4)	24 (64.9)	0.968
Warfarin; *n* (%)	2 (4.4)	5 (13.5)	0.235
Direct oral anticoagulants; *n* (%)	7 (15.6)	10 (27)	0.202
ACE inhibitor; *n* (%)	28 (62.2)	26 (70.3)	0.444
Angiotensin receptor blockers; *n* (%)	8 (17.8)	5 (13.5)	0.599
Statins; *n* (%)	45 (100)	36 (97.3)	0.451
Mineralocorticoid receptor antagonists; *n* (%)	16 (35.6)	21 (56.8)	0.055
Loop diuretics; *n* (%)	34 (75.6)	30 (81.1)	0.547
Iron supplements; *n* (%)	6 (13.3)	8 (21.6)	0.321

ACE—angiotensin-converting enzyme; RAAS—renin–angiotensin–aldosterone system. Data are presented in absolute and relative values: *n* (%).

**Table 5 life-15-00387-t005:** Results of logistic regression analysis.

Parameters	B (Regression Coefficient)	Wald Significance Test	P Level
Bilateral leg edema	−0.911	1.192	0.275
History of MI	0.174	0.058	0.810
NYHA class	0.493	0.830	0.362
Adherence to treatment based on Morisky–Green scores	−1.010	8.671	0.003
Total cholesterol, mmol/L	−0.104	0.831	0.362
Triglycerides, mmol/L	0.014	0.077	0.782
GDF-15, pg/mL	0.001	5.928	0.015
LVEF, %	−0.019	0.533	0.465
Mineralocorticoid receptor antagonists before hospitalization	−0.195	0.080	0.777
NTproBNP, pg/mL	0.000	1.907	0.167
Constant	0.875	0.116	0.734

MI—myocardial infarction; GDF-15—growth/differentiation factor 15; LVEF—left ventricular ejection fraction.

**Table 6 life-15-00387-t006:** Results of Cox regression.

Parameters	B (Regression Coefficient)	P Level	OR	95.0% CI for OR	
				Lower	Upper
Bilateral leg edema (1—yes; 0—no)	0.036	0.953	1.037	0.309	3.472
MI (0—no; 1—yes)	−0.282	0.592	0.755	0.270	2.112
Grade of HF by NYHA		0.946			
Grade of HF by NYHA(1)	−0.170	0.989	0.983	0.096	10.107
Grade of HF by NYHA(2)	−0.595	0.598	0.552	0.060	5.047
Grade of HF by NYHA(3)	−0.217	0.646	0.805	0.319	2.030
Total cholesterol, mmol/L	−0.196	0.042	0.822	0.680	0.993
LVEF %	0.003	0.872	1.003	0.967	1.041
Mineralocorticoid receptor antagonists	0.601	0.188	0.548	0.224	1.340
GDF-15 more than 2064	−1.872	0.001	0.154	0.050	0.474
Adherence to treatment (Morisky–Green scores)	−0.169	0.368	0.844	0.584	1.221
Triglycerides, mmol/L	0.001	0.797	1.001	0.992	1.010
NTproBNP, pg/mL	0.000	0.749	1.000	1.000	1.000

OR—odds ratio; CI—confidence interval; MI—myocardial infarction; HF—heart failure; LVEF—left ventricular ejection fraction.

## Data Availability

Data are available on reasonable request. The data underlying this article cannot be shared publicly to protect the privacy of individuals that participated in the study. The anonymized data may be shared on reasonable request to the corresponding author.

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
