# Peer review of "Predictors of Adverse Cardiovascular Events After CABG in Patients with Previous Heart Failure"

_life, 2025, doi:10.3390/life15030387_

Round 1

Reviewer 1 Report

Comments and Suggestions for Authors

Authors of this prospective observational study analyzed predictors for adverse events after CABG in patients with heart failure. Study and findings are interesting, but I have some comments. 

  1. The title implies that authors tried to define predictors for heart failure after CABG, but the aim of the study was to define predictors for adverse cardiovascular events in patients who had HF before CABG. So I think that the title should be modified.

  2. Why did authors exclude patients with MI? Did they exclude only patients with acute MI or patients with prior MI as well?

  3. Why did authors use logistic regression model first and then Cox regression model only for defining the risk for GDF-15?

  4. How do they choose which variables should be included in the regression model?

  5. Also, the study included a small number of patients.

  6. There are typo errors.

Comments on the Quality of English Language

The English could be improved. Also, there are some typo errors. 

Author Response

The authors thank the respected reviewer for the deep analysis of our work and valuable comments!

All necessary changes have been made to the article and highlighted in yellow.

  1. The title implies that authors tried to define predictors for heart failure after CABG, but the aim of the study was to define predictors for adverse cardiovascular events in patients who had HF before CABG. So I think that the title should be modified.

We have made changes to the title in accordance with the reviewer's recommendation.  A new title: «Predictors of adverse cardiovascular events after CABG in patients with previous heart failure»

  1. Why did authors exclude patients with MI? Did they exclude only patients with acute MI or patients with prior MI as well?

We excluded patients with MI that occurred within 6 months before inclusion in the study. Patients with MI more than 6 months ago were included in the study.

We have made the necessary clarifications in the text of the article.

«Non-inclusion criteria were: refusal to participate in the study, myocardial infarction (MI) within the last 6 months, stroke within the last 6 months….»

  1. Why did authors use logistic regression model first and then Cox regression model only for defining the risk for GDF-15?

Thank you for your question!

We used the logistic regression model at the initial stage to determine independent predictors of adverse cardiovascular events without taking into account the time after CABG when MACE developed. The next step was the cox regression, which confirmed the independent prognostic value of GDF15 even taking into account the time to endpoint parameter. However, the prognostic role of adherence to treatment was confirmed only in the logistic regression model and there was no statistical significance of this parameter in the cox regression. Therefore, both of these regression methods are presented in the article. Unless the esteemed reviewer insists otherwise, we would like to retain both of these analyses.

  1. How do they choose which variables should be included in the regression model?

We have added an explanation to the article:

Parameters that were significantly different between groups in univariate analysis were included in the multivariate logistic regression and Cox regression analysis. Transthoracic echocardiography parameters were significantly correlated with each other, and therefore we included only the left ventricular ejection fraction parameter in the regression analysis.

  1. Also, the study included a small number of patients.

Thanks for the comment, we have added a section "Study Limitations" to the article

« The main limitation of this study was the small sample size, but the advantage is that there were no patients lost to follow-up. Another limitation is the single-center nature of the study and pandemic COVID-19, which occurred during the follow-up period. However, the data from this work add the data on the important prognostic role of GDF-15 in patients with heart failure».

Reviewer 2 Report

Comments and Suggestions for Authors

I have read the article titled "Predictors of adverse cardiovascular events in patients with heart failure after CABG" sent to me for evaluation. First of all, I congratulate the authors for their efforts. My comments, criticisms and suggestions are listed below:

1. There are grammatical and spelling errors throughout the article. It should be reviewed.

2. The number of cases in the study seemed insufficient to me for such a subject. Since many parameters were analyzed, it seems a little difficult to obtain a meaningful result

3. The statement "The median time before the development of the MACE in the group with a GDF15 value less than 2064 pg/ml was 64 (50;80) month" is included in the Abstract section. However, the statement "The median follow-up period was 36 (26;43) months" is included in the Results section. The maximum follow-up periods in these two statements are inconsistent.

4. The statement "History of myocardial infarction, zones of hypo- or akinesia were registered more often in Group 2." is included in the Subgroup Comparison section. However, according to supplemental table 1, zones of hypo- or akinesia were registered more often in Group 1 [21 (46.7) in group 1 vs 9 (24.3) in group 2]

5. How did the p values ​​become so significant when the MVreg, grade and TVreg, grade degrees were the same in both groups? [MVreg, grade 1 (0.5;1) 1 (1;2) 0.003], [TVreg, grade 0 (0;0.875) 0 (0;1) 0.023]

6. The abbreviation SPLA, mmHg does not mean.

7. The LVEF% average for the first group was given as 60 (44;64). Considering that 35% of the patients in this group had LVEF below 50%, it would be good to review the group average LVEF value and min max values.

8. Although there is a nearly 2-fold difference between the NT-proBNP values ​​between the two groups [167.4(113.85;422.25) 326.7 (139;590)], it was not included in the multivariate analysis. It should be reviewed.

9. What does Interruptions; n (%) mean in Table 1?

10. Edema grading criteria should be given.

I think the issues I listed above should be reviewed again.

Best regards

Comments on the Quality of English Language

The English could be improved to more clearly express the research.

Author Response

The authors thank the respected reviewer for the deep analysis of our work, valuable comments, and his equitable assessment!

All necessary changes have been made to the article and highlighted in grey.

  1. There are grammatical and spelling errors throughout the article. It should be reviewed.

Thank you for the deep analysis of our paper, the article has been reviewed for grammatical and spelling errors.

  1. The number of cases in the study seemed insufficient to me for such a subject. Since many parameters were analyzed, it seems a little difficult to obtain a meaningful result

Thank you for the comment, we have added a section "Study Limitations" to the article

«The main limitation of this study was the small sample size, but the advantage is that there were no patients lost to follow-up. Another limitation is the single-center nature of the study and pandemic COVID-19, which occurred during the follow-up period. However, the data from this work add the data on the important prognostic role of GDF-15 in patients with heart failure».

  1. The statement "The median time before the development of the MACE in the group with a GDF15 value less than 2064 pg/ml was 64 (50;80) month" is included in the Abstract section. However, the statement "The median follow-up period was 36 (26;43) months" is included in the Results section. The maximum follow-up periods in these two statements are inconsistent.

Thank you for the clarification. We have added to the text the clarification that the median time before the development of the MACE in the group with a GDF15 value less than 2064 pg/ml was 64 (50;80) months and was predicted based on the Kaplan-Meier analysis.

«The median time before the development of the MACE which was predicted based on the Kaplan-Meier analysis in the group with a GDF-15 value less than 2064 pg/ml was 64 (50;80) months, in the group with a GDF-15 value more than or equal to 2064 pg/ml was 40 (34;46) months (p<0.001)»

  1. The statement "History of myocardial infarction, zones of hypo- or akinesia were registered more often in Group 2." is included in the Subgroup Comparison section. However, according to supplemental table 1, zones of hypo- or akinesia were registered more often in Group 1 [21 (46.7) in group 1 vs 9 (24.3) in group 2]

Thank you very much for finding this discrepancy. The text has been corrected.

«History of myocardial infarction, were registered more often in Group 2.» 

  1. How did the p values become so significant when the MVreg, grade and TVreg, grade degrees were the same in both groups? [MVreg, grade 1 (0.5;1) 1 (1;2) 0.003], [TVreg, grade 0 (0;0.875) 0 (0;1) 0.023].

We checked the data again, and indeed the developments in the degree of mitral and tricuspid regurgitation were statistically significant.

  1. The abbreviation SPLA, mmHg does not mean.

We have added data: “SPLA - systolic pressure pulmonary artery”

  1. The LVEF% average for the first group was given as 60 (44;64). Considering that 35% of the patients in this group had LVEF below 50%, it would be good to review the group average LVEF value and min max values.

Thank you. Data are given as median and interquartile range (Q25; Q75). The minimum and maximum values ​​of LVEF were 17% and 72%

We have added an explanation to tables 1-4:

Continuous variables were presented as median and interquartile range (Me (Q25; Q75). Categorical data were presented in absolute and relative values: n (%).

  1. Although there is a nearly 2-fold difference between the NT-proBNP values between the two groups [167.4(113.85;422.25) 326.7 (139;590)], it was not included in the multivariate analysis. It should be reviewed.

Thank you for your comment. Although the median NT value varied significantly, its interquartile range did not differ so much between the groups. Following your recommendation, we added NTproBNP to the logistic regression model and the cox model. NTproBNP was not an independent predictor of MACE in our cohort.

The tables 5 and 6, figures 1 and 4 were supplemented.

  1. What does Interruptions; n (%) mean in Table 1?

By the term «Interruptions; n (%)» we meant the number of patients who had experienced heart palpitations.

Replaced with the term «Heart palpitations, n (%)»

  1. Edema grading criteria should be given.

Thank you, additions have been made to the table 1:

«Bilateral leg edema, n (%)»

The criterion for bilateral leg edema was the appearance of pits that remains in the edematous area after pressure is applied on both legs.

Reviewer 3 Report

Comments and Suggestions for Authors

Please rephrase the last sentence in the Abstract part: “However, more studies are necessary before the implementation of GDF-15 in clinical practice can be recommended.”  That marker is used in clinical practice but not in the prognostic purposes that authors have investigated.

In the Introduction part please rephrase this part of the paragraph: “The prognosis in HF patients after CABG has been studied mainly in those with reduced left ventricular ejection fraction (LVEF). Current evidence showed a greater median survival of 7.73 years in HFrEF patients in the CABG group compared to those receiving optimal guideline directed HF therapy in whom the survival was 6.29 years….”

Were those patients with HFrEF with multivessel disease divided on those with and without CABG surgery? Please explain it better.

In the last sentence of the Introduction part where authors explain the purpose of the study, please clearly state were patients with HFrEF and CABG included or only those after CABG surgery with or without HFrEF. Be precise, thank you.  Also, number all biomarkers investigated as prognostic factors when explaining the purpose of the study.

In the part Material and Methods and subheading “End-points and follow up”, please explain what worsening of HF means (rehospitalization or use of parenteral diuretics…).

For the Results part: when laboratory analyses, other that for investigated biomarkers, and clinical examination were performed? Was that one day before surgery? When anamnesis was taken…. add that information in the Material and Methods part. In the Table 1 authors investigated the functional class of CAD, was that angina pectoris functional class?  

The text for so called “Predictor of outcomes” seems redundant. I suggest insert all information in the Table 5.

Discussion should be more comprehensive and concise.

Authors should add Study limitations part to this manuscript (number of patients, Covid pandemic, access to healthcare services etc.).

Comments on the Quality of English Language

should be improved

Author Response

The authors thank the respected reviewer for the deep analysis of our work, valuable comments, and his equitable assessment!

All necessary changes have been made to the article and highlighted in green.

  1. Please rephrase the last sentence in the Abstract part: “However, more studies are necessary before the implementation of GDF-15 in clinical practice can be recommended.” That marker is used in clinical practice but not in the prognostic purposes that authors have investigated.

Thank you very much, we have rephrased the last sentence in the Abstract:

However, further studies are needed to use GDF-15 as a prognostic marker in real-life clinical practice.

  1. In the Introduction part please rephrase this part of the paragraph: “The prognosis in HF patients after CABG has been studied mainly in those with reduced left ventricular ejection fraction (LVEF). Current evidence showed a greater median survival of 7.73 years in HFrEF patients in the CABG group compared to those receiving optimal guideline directed HF therapy in whom the survival was 6.29 years….”

Thank you very much! The introduction was changed  according to the reviewers' comments.

  1. Were those patients with HFrEF with multivessel disease divided on those with and without CABG surgery? Please explain it better.

All patients in our study underwent CABG. The necessary explanations have been included in the text.

A prospective observational study included consecutively hospitalized patients with HF who underwent CABG (inclusion period 2018–2020).

  1. In the last sentence of the Introduction part where authors explain the purpose of the study, please clearly state were patients with HFrEF and CABG included or only those after CABG surgery with or without HFrEF. Also, number all biomarkers investigated as prognostic factors when explaining the purpose of the study.

The sentence was rephrased:

The purpose of this study was to identify predictors of adverse cardiovascular events after CABG in patients with previous heart failure by looking at clinical data, standard laboratory and instrumental data, and biomarkers (NGAL, GDF-15, NTproBNP, TGF beta 1, hsCRP) and patients adherence before CABG.

  1. In the part Material and Methods and subheading “End-points and follow up”, please explain what worsening of HF means (rehospitalization or use of parenteral diuretics…).

Thanks. We have made a clarification.

Endpoints and follow-up. The primary endpoint of this study was a major adverse cardiac event (MACE), defined as cardiovascular death, acute ischemic events requiring unplanned revascularization, HF hospitalization, or stroke

  1. For the Results part: when laboratory analyses, other that for investigated biomarkers, and clinical examination were performed? Was that one day before surgery? When anamnesis was taken…. add that information in the Material and Methods part.

We have made a clarification.

Anamnesis, physical examination, standard laboratory and instrumental examination were performed 3-5 days before surgery.

  1. In the Table 1 authors investigated the functional class of CAD, was that angina pectoris functional class?

Yes, have made a clarification in table 1.

  1. The text for so called “Predictor of outcomes” seems redundant. I suggest insert all information in the Table 5.

Thanks, we have shortened the section “Predictor of outcomes”

  1. Discussion should be more comprehensive and concise.

Authors should add Study limitations part to this manuscript (number of patients, Covid pandemic, access to healthcare services etc.)

Thank you, we have changed the Discussion and added a " Study limitations "

«The main limitation of this study was the small sample size, but the advantage is that there were no patients lost to follow-up. Another limitation is the single-center nature of the study and pandemic COVID19, that occurred during the follow-up period. However, the data from this work add the data on the important prognostic role of GDF15 in patients with heart failure».

Reviewer 4 Report

Comments and Suggestions for Authors

The subject of the article is interesting but it contains many statements that are not very clear in the introduction and create a series of confusions.

I have inserted some of my comments directly into the text.

At line 63 - the cited guideline is old, from 2016

This is the new Guideline: 2021 ESC Guidelines for the diagnosis and treatment of acute and chronic heart failure: Developed by the Task Force for the diagnosis and treatment of acute and chronic heart failure of the European Society of Cardiology (ESC) With the special contribution of the Heart Failure Association (HFA) of the ESC.

Many variables were taken into account but these were no longer discussed and are no longer found in the conclusions.

Conclusions refers only to Higher GDF15 values and poor adherence to treatment but there a lot of factors related with the evolution after CABG… such as EF of LV, the presence of atrial fibrillation  etc.

Many references are old and with no direct relation with the article.

The subject of the article is interesting but it contains many statements that are not very clear in the introduction and create a series of confusions.

I have inserted some of my comments directly into the text.

At line 63 - the cited guideline is old, from 2016

This is the new Guideline: 2021 ESC Guidelines for the diagnosis and treatment of acute and chronic heart failure: Developed by the Task Force for the diagnosis and treatment of acute and chronic heart failure of the European Society of Cardiology (ESC) With the special contribution of the Heart Failure Association (HFA) of the ESC.

Many variables were taken into account but these were no longer discussed and are no longer found in the conclusions.

Conclusions refers only to Higher GDF15 values and poor adherence to treatment but there a lot of factors related with the evolution after CABG… such as EF of LV, the presence of atrial fibrillation  etc.

Many references are old and with no direct relation with the article.

I sugest to reformulate the introduction, the discussion section and the conclusions

Comments on the Quality of English Language

Minor English revisions are needed

Author Response

The authors thank the respected reviewer for the deep analysis of our work, valuable comments, and his equitable assessment!

All necessary changes have been made to the article and highlighted.

Thank you very much! We have completely reworked the introduction, supplemented the discussion and we have added a section "Study Limitations", according to your recommendations.

We also used your comments to make the necessary corrections to the article.

The list of references has also been revised.

Sincerely,

Authors

Round 2

Reviewer 1 Report

Comments and Suggestions for Authors

Authors significantly improved their manuscript. I have no further comments.

Thank you

Author Response

Comment 1:  Authors significantly improved their manuscript. I have no further comments.

Response 1: The authors thank the Reviewer for the scrupulous analysis of our manuscript. Your comments helped us significantly to improve the manuscript and describe our data more accurately. Thank you for the high assessment of our work and for your support.

Sincerely,

the Authors

Reviewer 2 Report

Comments and Suggestions for Authors

I have evaluated the new revised version of the article and the corrections made by the authors. I do not think that the article in its current form will make a significant contribution to the literature. My thoughts on the design of the study and the analysis of the data remain.

Best regards

Comments on the Quality of English Language

The English could be improved to more clearly express the research.

Author Response

Comment 1:  I have evaluated the new revised version of the article and the corrections made by the authors. I do not think that the article in its current form will make a significant contribution to the literature. My thoughts on the design of the study and the analysis of the data remain.

Response 1:  The authors thank the Reviewer for the scrupulous analysis of our manuscript. Your previous comments helped us to significantly improve the manuscript and describe our data more accurately. We appreciate your guidance and suggestions, which have helped us further refine the article.

You are right, our article has a number of limitations, but our data can be used to justify the necessity of larger studies needed to prove the prognostic role of GDF15 in HF patients undergone CABG. Our research group is studying prognosis and management of such patients and this is indeed a very difficult task.

Authors thank you for your advice and comments!  By using them we can refine the design of our further studies and hope to meet your high requirements in the future.

Thank you for your valuable opinion, help and cooperation!

Sincerely,

the Authors

Reviewer 3 Report

Comments and Suggestions for Authors

Authors significantly improved manuscript. All suggestions were accepted, all issues corrected. I do not see any major issues now and I suggest publishing.

Author Response

Comment 1:  Authors significantly improved manuscript. All suggestions were accepted, all issues corrected. I do not see any major issues now and I suggest publishing.

Response 1: The authors thank the Reviewer for the scrupulous analysis of our manuscript. Your comments helped us to significantly improve the manuscript and describe our data more accurately. Thank you for the high assessment of our work and for your support.

Sincerely,

the Authors

Reviewer 4 Report

Comments and Suggestions for Authors

I have read the new version of the article.

It was improved but still persist some unclear things such as lack of correlations between NT proBNP and the risk of MACE after CABG

I have inserted some comments directly into the text.

I suggest to re-write references.

Author Response

Comment 1:  It was improved but still persist some unclear things such as lack of correlations between NT proBNP and the risk of MACE after CABG

Response 1: Indeed, we did not obtain an association of NTproBNP and MACE in our study, which is inconsistent with the results of some other studies.  We think that this could be due to the fact that in our cohort the median NTproBNP values ​​were 167.4 pg/mL in patients without MACE and 326.7 pg/mL in patients with MACE. Whereas according to previous studies, higher NTproBNP values ​​(mainly more than 400 pg/mL) were associated with MACE [Comanici, M. et al., 2023]. We have included this explanation in the Discussion.

This is most likely due to the fact that in our cohort, the median NTproBNP values ​​were 167.4 pg/mL in patients without MACE and 326.7 pg/mL in patients with MACE. Whereas according to previous studies, higher NTproBNP values ​​(mainly more than 400 pg/mL) were associated with MACE [12].

Comment 2:  I have inserted some comments directly into the text.

Response 2: Thank you for your comments in the text. Based on your valuable feedback, we have made appropriate revisions to the text of article to more accurately reflect our results. New changes in the text are highlighted in pink.

Comment 3: I suggest to re-write references.

Response 3: Thank you for your comment. We have corrected the references according to your comments and the journals requirements.

Sincerely,

the Authors
